# Research on Sustainable Supplier Selection Based on the Rough DEMATEL and FVIKOR Methods

**Jing Zhang** [1], **Dong Yang** [1], **Qiang Li** [1], **Benjamin Lev** [2] and **Yanfang Ma** [1,*]

1   School of Economics and Management, Hebei University of Technology, Tianjin 300401, China; zhangjing@hebut.edu.cn (J.Z.); 201831705006@hebut.edu.cn (D.Y.); 202021702018@hebut.edu.cn (Q.L.)
2   LeBow College of Business, Drexel University, Philadephia, PA 19104, USA; bl355@drexel.edu
*   Correspondence: mayanfang@hebut.edu.cn

**Abstract:** In competitive global markets, sustainable suppliers are critical success factors for sustainable supply chain operations. Sustainable supplier selection must be based on a complex network of numerous indicators and experts' fuzzy linguistic terms. Considering the correlation between the evaluation criteria and the ambiguity of the criteria values, this paper proposes combining the rough DEMATEL method and the fuzzy VIKOR (FVIKOR) method to solve sustainable supplier selection problem. We determine 15 sustainable supplier evaluation criteria from economic, environmental and social dimensions. We also apply the rough DEMATEL method to determine the weight of evaluation indicators that are interrelated or even conflicting and use the FVIKOR method to determine supplier rankings by converting the fuzzy linguistic terms into precise information. The practicability of the proposed method is verified by an example of sustainable supplier selection.

**Keywords:** sustainable supplier selection; multi-criteria evaluation; rough DEMATEL method; fuzzy VIKOR method

## 1. Introduction

In competitive global markets, the supply chain of multiple relationships between upstream suppliers and downstream producers constitutes a new type of economic organization because it generates a fusion effect of core competences and resources from different organizations. Thus, sustainable suppliers are critical success factors for sustainable supply chain operations [1]. It is difficult for a separate organization, especially a downstream producer, to survive by depending on its own power [2]. The cost of raw materials and components account for a large proportion of the total production cost, close to 70%. Therefore, the selection of appropriate suppliers around the world is a critical issue that helps producers develop and enhance cost advantages in fiercely competitive global markets. Increased global awareness and enhanced government legislation on environmental effects pose challenges for multinational companies seeking low-cost suppliers in developing countries lacking environmental regulations [3]. Companies in developed countries must use the optimal measures to select multinational suppliers according to the economic and environmental objectives [4]. Multinational companies incorporated environmental protection and resource-saving consciousness into supplier management, and scholars focused on researching supplier evaluation of green procurement from different perspectives. Green supplier selection research has further expanded the research of traditional supplier evaluation and selection. In addition to considering the traditional economic performance indicators, studies have also focused on environmental performance indicators [5]. Through the supervision and control of suppliers' environmental protection measures, the overall economic and environmental benefits of the supply chain can be improved. However, in truth, multinational companies have to abandon incompetent suppliers who have difficulty meeting or exceeding green expectations due to limitations

in capacity, quality or service with the continuous improvement of environmental institutions in developing countries. Strict green measures lead to serious social problems, such as increasing the unemployment rate and decreasing the exports of suppliers, that adversely affect the competitiveness of multinational companies [6]. Research on green supplier evaluation ignores the impact of social factors, so the research trend has gradually shifted to sustainable supplier selection [7]. Building a comprehensive sustainable supplier evaluation criteria system is an important element of our research. As a multi-criteria evaluation problem, the evaluation index system built up by three dimensions of economy, environment and society is a complex network, and numerous indicators are bound to cause mutual relations and even conflicts [8]. Furthermore, evaluation values are often based on experts' linguistic terms to increase the potential profit and avoid decisional risk [9].

However, existing evaluation methods often ignore the correlations between indicators and the fuzziness of expert expression preferences. The purpose of this study is to propose a comprehensive sustainable supplier evaluation method, and compare this method with the AHP-VIKOR method proposed by Luthra et al. [10]. This method in our paper considers the interrelationship between different evaluation indexes and the fuzziness of external environmental information to determine the index weights and apply them to the selection of sustainable suppliers. First, the evaluation criteria of sustainable chain supplier selection are determined. Second, this paper uses a combination of the rough decision-making trial and evaluation laboratory (DEMATEL) and fuzzy vIsekriterijumska optimizacija i kompromisno resenje (FVIKOR) methods to evaluate and select sustainable suppliers. The DEMATEL method can effectively solve the interrelated effects of criteria and is applicable to situations where there are conflicts between criteria; thus, it is a precise choice for determining the weight of the criteria here, while rough set can effectively analyze the inconsistency and incompleteness of data. Therefore, we use a combination of DEMATEL and rough set theory to evaluate the index weight. The VIKOR method does not require a pairwise comparison and the calculation process is simple. In addition, due to the ambiguity of external environmental information and the expert expression preferences, decision makers can usually provide only linguistic assessments rather than accurate assessments. Therefore, we use the FVIKOR method to rank alternative suppliers.

The main contributions of this paper are as follows:

1. A comprehensive sustainable supplier evaluation index system is constructed.
2. To consider the issue of the mutual influence of evaluation criteria, an integrated evaluation selection method for sustainable suppliers is proposed.
3. The problem of expert expression preference is considered, and fuzzy theory is introduced into the multi-criteria evaluation framework.

The structure of this paper is as follows: Section 2 reviews the existing related literature. Section 3 details the approach presented in this article. Section 4 analyzes specific cases. Section 5 is a comparative analysis with other methods. Section 6 presents the conclusion, limitations and future prospects.

## 2. Literature Review

Research on supplier selection has basically focused on establishing an evaluation index system of suppliers and selecting evaluation methods. We reviewed the research status of these two research streams. Studies of supplier evaluation and selection systems have focused mainly on economic performance indicators and environmental performance indicators, and ignored the impact of social performance indicators [11]. Simultaneously, the method of supplier evaluation has become more complicated with the increasing number of indicators included in the supplier evaluation index system.

### 2.1. Sustainable Supplier Evaluation Index System

The traditional system is built on economic criteria, Dickson [12] systematically studied the construction of a supplier evaluation index system. After an analysis of 170 valid

questionnaires, 23 criteria for evaluating supplier performance were determined, and quality, delivery, and service were the most important criteria. Weber et al. [13], by summarizing 74 related articles, ranked the importance of evaluation criteria through statistical analysis and studied the importance of Just-In-Time (JIT) supplier evaluation criteria such as quality, delivery, price, production facilities, and production capacity as the essential criteria. Smytka and Clemens [14] summarized the experience of the total cost supplier selection method for General Electric Wiring Devices, considered risk factors, and divided the supplier evaluation index into risk factors, business demand factors, and measurable cost factors. With the worsening ecological environment, governments have formulated different environmental protection policies that force enterprises to consider the influence of environmental factors in the production process and introduce environmental criteria into the evaluation index system of suppliers. Noci [15] used environmental factors as the key factors in supplier performance evaluation and proposed four primary criteria as well 13 secondary criteria. Handfield et al. [16] used the InterContinental Hotels Group as an example to demonstrate that incorporating environmental factors into the procurement process would not impair purchasing power and proposed the 10 most important environmental performance criteria, including public disclosure of environmental records, ISO14000 certification, reverse logistics projects, and hazardous emissions management. Hsu et al. [17] proposed criteria such as carbon emission control, carbon-related management training, carbon information management system, carbon emission disclosure and reporting.

Currently, powerful pressures from stakeholders, consumers, nongovernmental organizations and local communities in supply chain management are increasing, enterprises shift the focus to the social issues, and the sustainability of supplier selection from a strategic perspective is becoming increasingly obvious [18]. Previous researches on supplier selection issues have not paid enough attention to social criteria [10]. Bai and Sarkis [19] summarized a series of sustainable supplier evaluation criteria from economic, environmental and social perspectives, including cost, pollution control, resource consumption, health and safety. Kannan et al. [20] proposed 11 primary criteria and 60 secondary criteria based on a literature review and affinity graph method. Govindan et al. [21] took into account the pressure of stakeholders and proposed a sustainable supplier evaluation index that includes cost, quality, delivery reliability, ecological design, local community impact, stakeholder impact, and employment practices. Song et al. [22] developed a sustainable supplier selection criteria system for solar air-conditioning manufacturers. In our research, we conducted a statistical analysis of economics, environmental, social performance criteria which is an important basis in the further analysis.

### 2.2. Sustainable Supplier Evaluation Method

The selection of suppliers is the key for manufacturing enterprises to reduce costs and increase competitiveness [23]. Vokurka et al. [24] proposed an expert system for evaluating and selecting alternative suppliers. This system combined with the strategic partnership of supplier selection can not only be used for supplier evaluation, but also serve as an interpretation tool, providing on-the-job training tools to help professionals become more proficient in a relatively short time. Patton [25] found that the traditional linear model is not widely used through the investigation of the actual industrial buyer's supplier selection problem and then proposed a manual judgment model. Chen et al. [26] proposed a multi-criteria decision making model (MCDM) based on fuzzy set theory that took into account the uncertainty degree of group decision problems, and fuzzy technique for order performance by similarity to ideal solution (FTOPSIS) method is used to sort candidate suppliers. Zhou et al. [27] considered the choice preference of suppliers, with cost and service quality as the main factors, and proposed a supplier choice preference model based on a hesitation fuzzy set.

Handfield et al. [16] considered that incorporating environmental factors into the procurement system would increase the complexity of the procurement process, so they

constructed a set of comprehensive information systems supporting environmental procurement with the analytical hierarchy process (AHP) method and evaluated suppliers. Kannan et al. [20] proposed a green supplier selection method based on fuzzy axiomatic design (FAD), which could not only select the best supplier but also determine the best alternative scheme. Zhao and Guo [28] conducted research on green supplier selection in thermal power plants and proposed a hybrid fuzzy multi-attribute decision-making method based on fuzzy entropy theory and the TOPSIS method. Guo et al. [29] studied the selection of green suppliers by apparel companies using fuzzy MCDM methods. Miranda-Ackerman et al. [30] studied the selection of green suppliers by using the MCDM multi-objective optimization strategy based on a genetic algorithm.

Supplier selection research took into account social and sustainability criteria, but the analysis and modeling of sustainable supply chains are still relatively new [31,32]. Bai and Sarkis [19] integrated gray system theory and rough set theory to incorporate sustainability attributes into the method based on the fuzzy inference system (FIS). Hendiani et al. [33] proposed a new MCDM based on interval type-2 trapezoidal fuzzy set under the triple bottom line criterion of sustainable supply chain management, which can accurately express subjective evaluations and qualitative assessments. Govindan et al. [21] considered the sustainability of supplier selection and proposed a supplier selection decision model based on the triple-bottom-line principle. Triangular fuzzy number is used to evaluate decision makers' preferences, and the fuzzy TOPSIS method is used to determine supplier rankings. Luthra et al. [10] proposed a comprehensive framework for sustainable supplier selection and evaluation using the AHP and VIKOR methods based on the Indian automotive industry, but did not take into account the ambiguity of information. Zhou et al. [34] established a benchmarking model based on uncertain dynamic data envelopment analysis (DEA), which can identify the poor performance period of each supplier. Sinha and Anand [35] proposed a MCDM model of a supplier selection framework for new product development based on environmental awareness through graph theory modeling. Their main contribution was to establish the degree of interrelationship between supplier selection attributes from the perspective of sustainability. However, subjectivity, imprecision and fuzziness continue to exist in the process of attribute assignment.

*2.3. Research Gaps and Highlights*

The above literature research indicates that various environmental factors and social factors have only begun to be integrated into the issue of supplier selection in recent years. As the dimension of sustainable supplier evaluation criteria increases, the supplier decision model becomes more complex. The supplier evaluation selection method is gradually transformed from a single algorithm to a combination of multiple algorithms. Nevertheless, there are still many shortcomings in existing supplier decision-making methods. For example, AHP is the method most commonly used to identify the index weight in supplier evaluation selection; it is a simple calculation that is easy for operators to apply. However, the premise of the AHP method is that the criteria are independent of each other, and the reciprocal influence between criteria is ignored. Analysis and study of the literature show that most evaluation and selection studies of sustainable suppliers do not consider correlations between criteria. However, there must be a correlation or conflict between economic, social and environmental criteria. Therefore, its main task is to create a complete evaluation index system of sustainable suppliers and improve the evaluation method of sustainable suppliers.

## 3. Methodology

There are many methods for supplier evaluation, such as the AHP method [16,36,37], TOPSIS method [21,38], VIKOR method [10,39], data envelopment analysis (DEA) method [40], and best-worst method (BWM) method [41]. Many of these methods do not consider the interrelation between criteria, while the DEMATEL method can analyze the causal relationship between factors. In addition, although the ANP approach also takes into account the

problem that criteria under different dimension affect each other, it considers that different dimensions are independent of each other. Our sustainable supplier evaluation index system can be divided into economic, environmental, and social three dimensions, inevitably there are interaction relationship between is not only dimension between internal indicators related problems, belonging to the various levels of that there is a connection between each evaluation index may also affect the relationship. DEMATEL method is concerned not only with the direct influence relationship between factors, but also considers the indirect influence relationship between all factors [42]. Therefore, we use the DEMATEL method pair to determine the weights of sustainable supplier performance indicators with complex network characteristics. However, the traditional DEMATEL method considers the criteria equally important when dealing with the relationship between criteria, which obviously does not conform to reality. Therefore, Song et al. [22] designed a new method, combining the DEMATEL method with rough set theory, to consider not only the interrelationship between different evaluation criteria but also the ambiguity and uncertainty of the information. The FVIKOR method is used to select suppliers. Compared with TOPSIS method, this method considers the distance between the decision scheme and the positive ideal solution and the distance from the decision scheme to the negative ideal solution, as well as the relative importance of these distances [43]. At the same time, due to the uncertainty of surrounding information in the process of evaluation decision and the limitations of the evaluation experts' individual knowledge, hesitant fuzzy sets better reflect the uncertainty of decision information and the language preferences of experts. Therefore, we combined the improved DEMATEL method with the FVIKOR method to propose a set supplier selection method. A schematic diagram of the proposed method is presented as Figure 1.

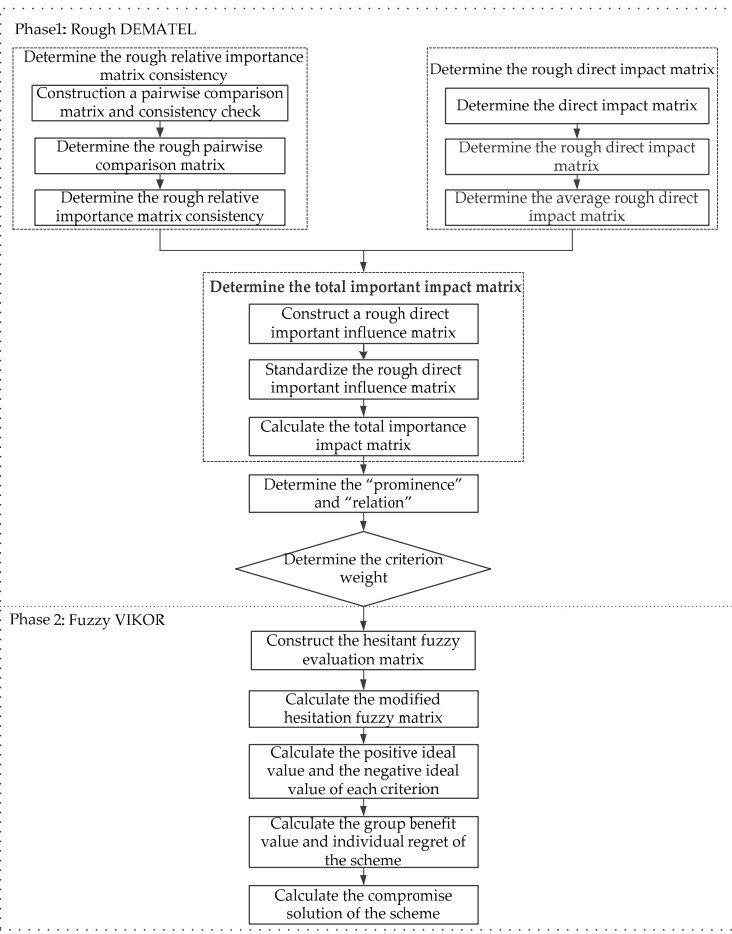

**Figure 1.** A schematic diagram of the proposed method.

*3.1. Preliminaries*

Rough set theory can effectively analyze data inconsistency and incompleteness. Its definition is as follows:

Set $U$ as the domain, and there are $m$ kinds of judgments, which are shown as $J = \left\{ S^1, S^2, \ldots, S^m \right\}$, and an orderly relationship, $S^1 < S^2 < \cdots < S^m$.

The individuals contained in the domain are $x$, $\forall X \subseteq U$, and the lower approximation set and upper approximation set of $X$ are $\underline{Apr}(S^i) = \cup\left\{ x \in U/J(X) \leq S^i \right\}$ and $\overline{Apr}(S^i) = \cup\left\{ x \in U/J(X) \geq S^i \right\}$.

The lower limit and upper limit of the rough interval of $X$ are $\underline{Lim}(S^i) = \left( \prod_{i=1}^{N_i} x_i \right)^{1/N_i}$ and $\overline{Lim}(S^i) = \left( \prod_{i=1}^{N_i} y_i \right)^{1/N_i}$ .where $x_i$ is the lower approximation of $S^i$ and $y_i$ is the upper approximation of $S^i$.

*3.2. Criteria Weight Determination Method*

We use the rough DEMATEL method to determine the criteria weight. First, experts provide the relative importance matrix and the direct impact matrix of each indicator, and we then use the rough set to convert the evaluation results into rough numbers. Then, the DEMATEL method is utilized to calculate the weight of each criterion. The specific steps are as follows:

Step 1: Determine the rough relative importance matrix.

Step 1.1: Construct a pairwise comparison matrix and consistency check.

First, the experts compare the relative importance of sustainable supplier criteria and obtain the evaluation matrix $R_k$. Second, the consistency check is performed on the pairwise comparison matrix. If it does not satisfy the consistency test, the expert adjusts the evaluation results.

$$R_k = \left[ R_{ij}^k \right]_{n \times n}, k = 1, 2, \ldots, m \tag{1}$$

where $r_{ij}^k$ represents the judgment of the *kth* expert on the relative importance of criterion $i$ and criterion $j$, $m$ presents the number of experts, and $n$ is the number of criteria.

$$CI = \frac{\chi_{\max} - n}{n - 1} \tag{2}$$

$$CR = \left( \frac{CI}{RI(n)} \right) \tag{3}$$

where is the consistency coefficient, $\chi_{\max}$ is the maximum eigenvalue of matrix $R_k$, $\beta$ is the consistency ratio, $n$ is the dimension of $R_k$ and $RI(n)$ is the random exponent. When the consistency ratio $CR < 0.1$, the pairwise comparison matrix passes the consistency test. When $CR > 0.1$, the decision makers need to adjust their evaluation results. The pairwise comparison matrix after the consistency check is $\widetilde{R} = \left[ r_{ij} \right]_{n \times n}$.

Step 1.2: Determine the rough pairwise comparison matrix.

Suppose $J = \left\{ r_{ij}^1, r_{ij}^2, \ldots, r_{ij}^m \right\}$ is a group of judgments of experts; then, the lower and upper approximations of $R_{ij}^k$ can be obtained by

$$\underline{Apr}(r_{ij}^k) = \cup\left\{ X \in U/J(A) \leq r_{ij}^k \right\} \tag{4}$$

$$\overline{Apr}(r_{ij}^k) = \cup\left\{ X \in U/J(A) \geq r_{ij}^k \right\} \tag{5}$$

where $\underline{Apr}(r_{ij}^k)$ is the lower approximation of $R_{ij}^k$, $\overline{Apr}(r_{ij}^k)$ is the upper approximation of $R_{ij}^k$, $U$ is a collection of all evaluation elements, and $X$ is any element of $U$.

Then, the evaluation sequence in the $\widetilde{R}$ matrix is converted into a rough number.

$$\underline{Lim}\left(r_{ij}^{k}\right) = \left(\prod_{m=1}^{N_{ijL}} x_{ij}\right)^{1/N_{ijL}} \tag{6}$$

$$\overline{Lim}\left(r_{ij}^{k}\right) = \left(\prod_{m=1}^{N_{ijL}} y_{ij}\right)^{1/N_{ijL}} \tag{7}$$

where $\underline{Lim}(r_{ij}^{k})$ is the lower limit of the rough number, $\overline{Lim}(r_{ij}^{k})$ is the upper limit of the rough number, and $x_{ij}$ and $y_{ij}$ are the lower approximation and the upper approximation, respectively. $N_{ijL}$ is the number of elements included in the upper middle approximation.

Then, the rough two-two matrix is $RN\left(r_{ij}^{k}\right)$, expressed as

$$RN(r_{ij}^{K}) = \left[\underline{Lim}(r_{ij}^{k}), \overline{Lim}(r_{ij}^{k})\right] = \left[r_{ij}^{kL}, r_{ij}^{kU}\right] \tag{8}$$

where $r_{ij}^{kL}$ and $r_{ij}^{kU}$ are the lower limit and upper limit of $RN\left(r_{ij}^{k}\right)$ of the *kth* matrix, respectively.

Then, using the geometric averaging method, the evaluation results of different experts are aggregated, and the judgment group $\widetilde{R}_{ij} = \left\{r_{ij}^{1}, r_{ij}^{2}, \ldots, r_{ij}^{m}\right\}$ of the *kth* expert is converted into the rough sequence $RN\left(r_{ij}^{k}\right)$.

$$RN(\widetilde{r_{ij}}) = \left\{\left[r_{ij}^{1L}, r_{ij}^{1U}\right], \left[r_{ij}^{2L}, r_{ij}^{2U}\right], \ldots, \left[r_{ij}^{mL}, r_{ij}^{mU}\right]\right\} \tag{9}$$

Step 1.3: Construct an average rough pairwise comparison matrix.

Let the average value of the rough numbers be $\overline{RN(\widetilde{r_{ij}})}$. The results are expressed as $\overline{RN(\widetilde{r_{ij}})} = \left[r_{ij}^{L}, r_{ij}^{U}\right]$.

$$r_{ij}^{L} = \left(\prod_{k=1}^{m} r_{ij}^{kL}\right)^{1/m} \tag{10}$$

$$r_{ij}^{U} = \left(\prod_{k=1}^{m} r_{ij}^{kU}\right)^{1/m} \tag{11}$$

Step 2: Determine the rough direct impact matrix.
Step 2.1: Determine the direct impact matrix.
Let the direct impact matrix of the *kth* expert be $M_k$.

$$M_k = \left[s_{ij}^{k}\right]_{n \times n} \tag{12}$$

where $\left[\widetilde{s_{ij}}\right]_{n \times n}$ represents the direct impact of criterion $C_i$ provided by the *kth* expert on criterion $C_j$.

Step 2.2: Determine the rough direct impact matrix.
Use Formulas (7)–(10) to organize the direct influence of the *kth* expert in a rough direct influence matrix $\widetilde{M}_k$.

$$\widetilde{M}_k = \left[\left[s_{ij}^{kL}, s_{ij}^{kU}\right]\right] \tag{13}$$

where $s_{ij}^{kL}$ and $s_{ij}^{kU}$ represent the lower and upper bounds of the rough interval form, respectively.

Step 2.3: Determine the average rough direct impact matrix $M$.
Calculate the average value matrix $\widetilde{M}_k$ to get the average rough direct impact matrix $M$.

$$M = \left[\left[s_{ij}^{L}, s_{ij}^{U}\right]\right]_{n \times n} \tag{14}$$

where $s_{ij}^L$ and $s_{ij}^U$ are the lower and upper limits of $\left[s_{ij}^L, s_{ij}^U\right]$, respectively.

$$s_{ij}^L = \left(\prod_{k=1}^{m} s_{ij}^{kL}\right)^{1/m} \tag{15}$$

$$s_{ij}^U = \left(\prod_{k=1}^{m} s_{ij}^{kU}\right)^{1/m} \tag{16}$$

Step 3: Determine the total important impact matrix.

Step 3.1: Construct a rough direct important influence matrix $D$.

$$D = R \times M = \left[d_{ij}\right]_{n \times n} \tag{17}$$

$$d_{ij} = \left[d_{ij}^L, d_{ij}^U\right] = \left[r_{ij}^L \times s_{ij}^L, r_{ij}^U \times s_{ij}^U\right] \tag{18}$$

where $R$ is the average rough pairwise comparison matrix and $M$ is the average rough direct influence matrix.

Step 3.2: Standardize the rough direct important influence matrix.

$$C = \left[\tilde{u}_{ij}\right]_{n \times n} \tag{19}$$

$$\tilde{u}_{ij} = \left[\frac{d_{ij}^L}{\gamma}, \frac{d_{ij}^U}{\gamma}\right] = \left[u_{ij}^L, u_{ij}^U\right] \tag{20}$$

$$\gamma = \max_{1 \le i \le n}\left(\sum_{j=1}^{n} d_{ij}^U\right) \tag{21}$$

where $C$ is a normalized matrix and $u_{ij}^L$ and $u_{ij}^U$ are the lower and upper limits of the rough interval, respectively.

Step 3.3: Calculate the total importance impact matrix.

Decompose the rough numbers in the normalized rough group directly into the matrix $C$ in separate submatrices $C^L$ and $C^U$.

$$C^L = \left[u_{ij}^L\right]_{n \times n} \ and \ C^U = \left[u_{ij}^U\right]_{n \times n} \tag{22}$$

Set the total importance matrix as $T^s (s = L, U)$, which is obtained by

$$T^L = \left[t_{ij}^L\right]_{n \times n} = C^L \left(1 - C^L\right)^{-1} \tag{23}$$

$$T^L = \left[t_{ij}^L\right]_{n \times n} = C^L \left(1 - C^L\right)^{-1} \tag{24}$$

The rough total importance matrix can be expressed as $T = \left[\tilde{t}_{ij}\right]_{n \times n}$, and the rough number is then converted to a certain value.

$$\tilde{Z}_i^L = \left(z_i^L - \min_i z_i^L\right) / \Delta_{\min}^{\max} \tag{25}$$

$$\tilde{Z}_i^U = \left(z_i^U - \min_i z_i^L\right) / \Delta_{\min}^{\max} \tag{26}$$

$$\Delta_{\min}^{\max} = \max_i z_i^U - \min_i z_i^L \tag{27}$$

where $z_i^L$ represents the lower limit of $\tilde{z}_i$, $z_i^U$ represents the upper limit of $\tilde{z}_i$, $\tilde{z}_i^L$ and $\tilde{z}_i^U$ represents the normalized forms of $z_i^L$ and $z_i^U$.

Determine the total normalized value.

$$\beta_i = \frac{\widetilde{z}_i^L \times (1 - \widetilde{z}_i^L) + \widetilde{z}_i^U \times \widetilde{z}_i^U}{1 - \widetilde{z}_i^L + \widetilde{z}_i^U} \tag{28}$$

Calculate the final fixed value $\widetilde{z}_i^{der}$ of $\widetilde{z}_i$ to obtain the rough total importance impact matrix $T^*$.

$$\widetilde{z}_i^{der} = \min_i z_i^L + \beta_i \Delta_{\min}^{\max} \tag{29}$$

$$T^* = \left[ t_{ij} \right]_{n \times n} \tag{30}$$

Step 3.4: Determine the "prominence" and "relation".

$x_i$ and $y_i$ represent the row sum and column sum of matrix $T^*$, respectively.

$$x_i = \sum_{j=1}^{n} t_{ij}, i = 1, 2, \ldots, n \tag{31}$$

$$y_j = \sum_{i=1}^{n} t_{ij}, j = 1, 2, \ldots, n \tag{32}$$

where $x_i$ represents the total effects, both direct and indirect, given by criterion $i$ to the other criteria and $y_i$ shows the total effects, both direct and indirect, received by criterion $j$ from the other criteria.

$$P_i = x_i + y_i, i = j \tag{33}$$

$$R_i = x_i - y_i, i = j \tag{34}$$

$$w_i = \frac{\sqrt{P_i^2 + R_i^2}}{\sum_{i=1}^{n} \sqrt{P_i^2 + R_i^2}} \tag{35}$$

"Prominence" is represented by $P_i$, "Relation" is represented by $R_i$, and $w_i$ is the composite weight of the *ith* criterion.

### 3.3. Supplier Selection Method

The VIKOR method is a MCDM method based on ideal points proposed by Opricovic. First, the criteria of each alternative scheme are ranked by calculating their proximity to the ideal value. When the stability requirement is satisfied, the selection is performed according to the ranking results of the scheme. At the same time, this method considers the subjective preferences of decision makers that are in line with reality and have been applied in many fields, such as manufacturing, banking, and pharmaceutical industries. The specific steps are as follows:

Step 1: Set up experts and decision teams.

Step 2: Develop evaluation criteria through a literature review and expert opinions.

Step 3: Create the criteria evaluation matrix $R = \left[ r_{ij} \right]_{n \times n}$.

Step 4: Calculate the weight of the criteria using the index weight determination method proposed in 3.2, consider the expert weight $\omega_k$ and calculate the index weight $w_i$.

Step 5: Use the FVIKOR method to evaluate the alternative scheme; expert $y_k$ provides the hesitant fuzzy evaluation information for each criterion $C_j$ of alternative scheme $A_i$.

Let the hesitation fuzzy evaluation matrix $V$ be

$$V = \begin{bmatrix} v_{11}^y & v_{12}^y & \cdots & v_{ij}^y \\ v_{21}^y & v_{22}^y & \cdots & \vdots \\ \vdots & \vdots & \ddots & \vdots \\ v_{i1}^y & v_{i2}^y & \cdots & v_{i1}^y \end{bmatrix} \tag{36}$$

The hesitant fuzzy set is expressed as $v_{ij}$.

$$v_{ij} = \{s_1, s_2 \cdots s_n\} \tag{37}$$

Due to the different cognition and preferences of experts, the number of hesitant fuzzy elements in different evaluation information provided by different experts for the same criteria of the same alternative may be different. The hesitation fuzzy set is expanded according to optimistic or pessimistic rules for creating its elements. The number is $l_{ij}$, and the expanded hesitation fuzzy matrix $D$ is obtained.

$$l_{ij} = \max\left\{l_{ij}^1, l_{ij}^2, \cdots, l_{ij}^p\right\} \tag{38}$$

$$D = \begin{bmatrix} d_{11}^y & d_{12}^y & \cdots & d_{ij}^y \\ d_{21}^y & d_{22}^y & \cdots & \vdots \\ \vdots & \vdots & \ddots & \vdots \\ d_{i1}^y & d_{i2}^y & \cdots & d_{i1}^y \end{bmatrix} \tag{39}$$

$$d_{ij} = \{e_1, e_2 \cdots e_l\} \tag{40}$$

Then we determine the modified hesitation fuzzy matrix $H = \left[h_{ij}\right]_{m \times n}$.

$$h_{ij} = \frac{\sum\limits_{i=1}^{l} e_i}{l} \tag{41}$$

Determine the positive ideal value and the negative ideal value of each criterion.

$$h_j^* = \max\left\{h_{ij}^y, i = 1, 2 \cdots q, j = 1, 2 \cdots n, y = 1, 2 \cdots k\right\} \tag{42}$$

$$h_j^- = \min\left\{h_{ij}^y, i = 1, 2 \cdots q, j = 1, 2 \cdots n, y = 1, 2 \cdots k\right\} \tag{43}$$

Calculate the group benefit value and individual regret degree of the scheme $R_{A_i}$.

$$S_{A_i} = \sum_{j=1}^{n} w_j \left[\frac{h_j^* - h_{ij}}{h_j^* - h_j^-}\right] \tag{44}$$

$$R_{A_i} = \max\left[w_j \left(\frac{h_j^* - h_{ij}}{h_j^* - h_j^-}\right)\right] \tag{45}$$

where $h_{ij}$ represents the revised hesitant fuzzy number, $h_j^*$ and $h_j^-$ represent the positive and negative ideal values of each criterion, $S_{A_i}$ represents the group benefit value of the *ith* alternative, and $R_{A_i}$ represents the personal regret degree of the *ith* alternative.

Then we determine the compromise solution of scheme $A_i$; the results are sorted by the size of the compromise solution, and the solution with the smallest $f_{A_i}$ value is selected.

$$f_{A_i} = V\frac{S_{A_i} - S^*}{S^- - S^*} + (1 - V)\frac{R_{A_i} - R^*}{R^- - R^*} \tag{46}$$

$$S^- = \max\nolimits_{A_i} S_{A_i} \tag{47}$$

$$S^* = min\nolimits_{A_i} S_{A_i} \tag{48}$$

$$R^- = \max\nolimits_{A_i} R_{A_i} \tag{49}$$

$$R^* = min\nolimits_{A_i} R_{A_i} \tag{50}$$

where $f_{A_i}$ represents the compromise solution for scheme $A_i$ and $V$ represents the maximum set utility weight for the criterion.

## 4. Case Study

We take a company as an example, and the case company hopes to reduce the negative impact of its business on the environment and society and to cultivate a green and sustainable development culture by incorporating sustainability into the assessment. Currently, the company managers have some difficulties in sustainable supplier assessment and supply chain management practices. As a result, the company is seeking a new way to evaluate suppliers to better achieve sustainability for its business. The managers hope to comprehensively consider the evaluation criteria and determine a reasonable and effective evaluation method. First, we developed a reasonable evaluation index system for sustainable suppliers through the literature review and expert opinions. Then, the company constructed a four-person decision-making team composed of experts and company professional managers to evaluate the criteria of the sustainable supplier evaluation index system and determine the weight of each criterion. The company's alternative suppliers are used to verify the effectiveness of the proposed method.

### 4.1. Select the Sustainable Supplier Evaluation Criteria and Determine the Criteria Weights

At this stage, a decision-making team composed of four experts (two experts and two senior managers from the procurement and production departments of the case company) was formed. Through the literature analysis and the opinions of experts, the three main dimensions and 15 evaluation criteria for sustainable supplier selection were determined. The case company had four candidate suppliers. First, the experts evaluated each index, created the pairwise comparison matrix of the criteria and the mutual influence matrix, and then used the rough DEMATEL method to calculate the weight of each criterion.

Step 1: Determine the rough relative importance matrix.

Step 1.1: Construct the pairwise comparison matrix and consistency test.

The relative importance of sustainable supplier standards was determined by the four experts using the 7-point scale (the scores of equally important, moderately important, strongly important, intermediate importance, moderately unimportant, strongly unimportant and intermediate unimportance are 1, 3, 5, 2(4), 1/3, 1/5 and 1/2(1/4) respectively); then, the scale was checked to determine whether the suppliers passed the consistency ratio shown in Table 1.

**Table 1.** The pairwise comparison matrix.

| | Costs | Quality | Delivery Reliability | ... | Local Community Influence | Social Responsibility Management System |
|---|---|---|---|---|---|---|
| Costs | 1, 1, 1, 1 | 1/2, 1/2, 1, 1/2 | 1, 2, 2, 2 | ... | 5, 4, 4, 3 | 1, 2, 1, 2 |
| Quality | 2, 2, 1, 2 | 1, 1, 1, 1 | 2, 2, 2, 2 | ... | 5, 4, 5, 5 | 2, 2, 2, 2 |
| Delivery reliability | 1, 1/2, 1/2, 1/2 | 1/2, 1/2, 1/2, 1/2 | 1, 1, 1, 1 | ... | 4, 3, 3, 4 | 1, 1, 1/2, 1 |
| Technology capability | 1/3, 1/3, 1/3, 1/3 | 1/3, 1/2, 1/3, 1/3 | 1/2, 1/2, 1/2, 1/2 | ... | 4, 2, 3, 2 | 1/2, 1/2, 1/2, 1/2 |
| Service | 1/2, 1/2, 1/2, 1/3 | 1/2, 1/3, 1/3, 1/4 | 1/2, 1, 1/2, 1 | ... | 3, 3, 2, 4 | 1/2, 1/2, 1/2, 1 |
| Financial situation | 1/4, 1/3, 1/3, 1/3 | 1/4, 1/4, 1/4, 1/4 | 1/3, 1/3, 1/3, 1/2 | ... | 1, 1, 1, 2 | 1/4, 1/3, 1/3,1/3 |
| Pollution production | 1, 1, 1/2, 1/2 | 1/2, 1/2, 1/3, 1/2 | 1, 2, 1, 1 | ... | 3, 3, 4, 3 | 1, 1, 1, 1 |
| Environmental management system | 1, 1, 1, 1 | 1/2, 1/2, 1/2, 1/2 | 2, 3, 2, 1 | ... | 3, 4, 3, 3 | 1, 1, 2, 2 |
| Green product | 1, 1/2, 1/3, 1/2 | 1/2, 1/3, 1/3, 1/3 | 1, 1/2, 1/2, 1/2 | ... | 1, 2, 2, 2 | 1/2, 1/2, 1/2, 1/2 |
| Pollution control | 1/2, 1/2, 1/2, 1/2 | 1/3, 1/2, 1/3, 1/4 | 1/2, 1, 1/2, 1/3 | ... | 2, 3, 3, 3 | 1/3, 1/2, 1/2, 1/2 |
| Green image | 1/3, 1/4, 1/3, 1/4 | 1/4, 1/5, 1/4, 1/5 | 1/2, 1/3, 1/3, 1/3 | ... | 1, 1, 1, 1 | 1/2, 1/3, 1/3, 1/3 |
| Health and safety | 1/2, 1, 1/2, 1/2 | 1/2, 1/2, 1/3, 1/3 | 2, 2, 1, 1 | ... | 4, 3, 3, 2 | 1, 1, 1/2, 1/2 |
| Contractual stakeholder influence | 1/4, 1/4, 1/3, 1/4 | 1/3, 1/5, 1/4, 1/4 | 1/3, 1/3, 1/3, 1/3 | ... | 1, 1, 1, 2 | 1/2, 1/3, 1/4, 1/3 |
| Local community influence | 1/5, 1/4, 1/4, 1/3 | 1/5, 1/4, 1/5, 1/5 | 1/4, 1/3, 1/3, 1/4 | ... | 1, 1, 1, 1 | 1/3, 1/3, 1/4, 1/3 |
| Social responsibility management system | 1, 1/2, 1, 1/2 | 1/2, 1/2, 1/2, 1/2 | 1, 1, 2, 1 | ... | 3, 3, 4, 3 | 1, 1, 1, 1 |

As mentioned before, there are 15 indicators, but some indicators are replaced by ellipsis ( ... ... ) in Table 1 because of the limitation of layout space. The following tables are similar.

Step 1.2: Determine the rough pairwise comparison matrix.

By using Equations (4)–(9), the numbers in matrix $S$ are reduced to rough numbers.

Step 1.3: Construct the average rough pairwise comparison matrix.

By using Equations (10)–(13), the mean rough pairwise comparison matrix is obtained, as shown in Table 2.

**Table 2.** The average rough pairwise comparison matrix.

| | Costs | Quality | Delivery Reliability | ... | Local Community Influence | Social Responsibility Management System |
|---|---|---|---|---|---|---|
| Costs | [1.000,1.000] | [0.522,0.677] | [1.477,1.915] | ... | [3.534,4.372] | [1.189,1.682] |
| Quality | [1.477,1.915] | [1.000,1.000] | [2.000,2.000] | ... | [4.535,4.931] | [2.000,2.000] |
| Delivery reliability | [0.522,0.677] | [0.500,0.500] | [1.000,1.000] | ... | [3.224,3.722] | [0.738,0.958] |
| Technology capability | [0.333,0.333] | [0.342,0.398] | [0.500,0.500] | ... | [2.216,3.130] | [0.500,0.500] |
| Service | [0.452,0.487] | [0.298,0.334] | [0.595,0.841] | ... | [2.515,3.357] | [0.522,0.677] |
| Financial situation | [0.294,0.327] | [0.250,0.250] | [0.342,0.398] | ... | [1.044,1.354] | [0.294,0.327] |
| Pollution production | [0.595,0.841] | [0.419,0.487] | [1.044,1.354] | ... | [3.054,3.402] | [1.000,1.000] |
| Environmental management system | [1.000,1.000] | [0.500,0.500] | [1.472,2.236] | ... | [3.054,3.402] | [1.189,1.682] |
| Green product | [0.430,0.680] | [0.342,0.398] | [0.522,0.677] | ... | [1.477,1.915] | [0.500,0.500] |
| Pollution control | [0.500,0.500] | [0.298,0.398] | [0.447,0.680] | ... | [2.512,2.925] | [0.419,0.487] |
| Green image | [0.269,0.310] | [0.211,0.236] | [0.342,0.398] | ... | [1.000,1.000] | [0.342,0.398] |
| Health and safety | [0.522,0.677] | [0.368,0.452] | [1.189,1.682] | ... | [2.515,3.357] | [0.595,0.481] |
| Contractual stakeholder influence | [0.255,0.284] | [0.229,0.283] | [0.333,0.333] | ... | [1.044,1.354] | [0.298,0.398] |
| Local community influence | [0.229,0.283] | [0.203,0.221] | [0.296,0.310] | ... | [1.000,1.000] | [0.294,0.327] |
| Social responsibility management system | [0.595,0.841] | [0.500,0.500] | [1.044,1.354] | ... | [3.054,3.402] | [1.000,1.000] |

Step 2: Determine the rough direct impact matrix.

Step 2.1: Construct the direct influence matrix.

The interaction between the different evaluation criteria of sustainable suppliers was evaluated by the four experts based on the linguistic terms (the scores of very high influence, high influence, medium influence, low influence and no influence are 4, 3, 2, 1, 0), and the corresponding scores are shown in Table 3.

**Table 3.** The direct influence matrix.

| | Costs | Quality | Delivery Reliability | ... | Local Community Influence | Social Responsibility Management System |
|---|---|---|---|---|---|---|
| Costs | 0, 0, 0, 0 | 4, 4, 4, 4 | 2, 2, 1, 2 | ... | 0, 0, 0, 0 | 0, 0, 0, 0 |
| Quality | 4, 4, 3, 4 | 0, 0, 0, 0 | 0, 1, 0, 0 | ... | 0, 0, 0, 1 | 0, 0, 0, 1 |
| Delivery reliability | 3, 2, 2, 3 | 0, 0, 0, 0 | 0, 1, 2, 0 | ... | 0, 0, 0, 0 | 0, 0, 0, 0 |
| Technology capability | 4, 3, 3, 3 | 4, 3, 3, 4 | 2, 1, 2, 2 | ... | 0, 0, 0, 0 | 1, 0, 0, 0 |
| Service | 0, 1, 1, 0 | 0, 0, 0, 0 | 2, 2, 2, 1 | ... | 0, 0, 0, 0 | 0, 0, 0, 0 |
| Financial situation | 2, 0, 0, 0 | 0, 0, 0, 0 | 1, 2, 2, 2 | ... | 1, 1, 3, 2 | 0, 1, 0, 0 |
| Pollution production | 2, 3, 2, 2 | 0, 0, 0, 0 | 0, 0, 0, 0 | ... | 4, 4, 4, 3 | 1, 2, 2, 2 |
| Environmental management system | 2, 2, 2, 3 | 0, 0, 0, 0 | 0, 0, 1, 0 | ... | 4, 2, 2, 3 | 2, 1, 1, 1 |
| Green product | 3, 4, 2, 3 | 1, 1, 2, 1 | 0, 0, 0, 0 | ... | 4, 1, 3, 3 | 1, 2, 1, 2 |
| Pollution control | 3, 3, 1, 2 | 0, 0, 0, 0 | 0, 0, 0, 0 | ... | 4, 3, 2, 3 | 2, 1, 2, 1 |
| Green image | 1, 2, 2, 1 | 1, 2, 1, 2 | 0, 0, 0, 0 | ... | 2, 2, 1, 2 | 1, 1, 1, 2 |
| Health and safety | 0, 0, 0, 0 | 0, 0, 0, 0 | 0, 0, 0, 0 | ... | 1, 0, 1, 1 | 3, 2, 2, 2 |
| Contractual stakeholder influence | 2, 1, 2, 2 | 1, 1, 1, 2 | 1, 1, 2, 1 | ... | 2, 2, 2, 2 | 2, 2, 1, 2 |
| Local community influence | 0, 0, 1, 0 | 0, 0, 0, 0 | 0, 0, 0, 0 | ... | 0, 0, 0, 0 | 3, 1, 2, 1 |
| Social responsibility management system | 2, 1, 2, 2 | 2, 1, 1, 1 | 0, 0, 0, 0 | ... | 2, 2, 1, 2 | 0, 0, 0, 0 |

Step 2.2: Determine the rough direct-influence matrix.

According to Equation (13), the fractions in Table 3 are reduced to rough numbers.

Step 2.3: Calculate the average rough direct influence matrix.

According to Equations (14)–(16), the average roughness directly affects matrix $M$, and the results are shown in Table 4.

**Table 4.** The average rough direct influence matrix.

| | Costs | Quality | Delivery Reliability | ... | Local Community Influence | Social Responsibility Management System |
|---|---|---|---|---|---|---|
| Costs | [0.000,0.000] | [4.000,4.000] | [1.477,1.915] | ... | [0.000,0.000] | [0.000,0.000] |
| Quality | [3.527,3.929] | [0.000,0.000] | [0.000,0.000] | ... | [0.000,0.000] | [0.000,0.000] |
| Delivery reliability | [2.213,2.711] | [0.000,0.000] | [0.000,0.000] | ... | [0.000,0.000] | [0.000,0.000] |
| Technology capability | [2.539,3.591] | [3.224,3.722] | [1.477,1.915] | ... | [0.000,0.000] | [0.000,0.000] |
| Service | [0.000,0.000] | [0.000,0.000] | [1.477,1.915] | ... | [0.000,0.000] | [0.000,0.000] |
| Financial situation | [0.000,0.000] | [0.000,0.000] | [1.477,1.915] | ... | [1.185,2.060] | [0.000,0.000] |
| Pollution production | [2.051,2.388] | [0.000,0.000] | [0.000,0.000] | ... | [3.527,3.929] | [1.477,1.915] |
| Environmental management system | [2.051,2.388] | [0.000,0.000] | [0.000,0.000] | ... | [2.216,3.130] | [1.044,1.354] |
| Green product | [2.515,3.357] | [1.044,1.354] | [0.000,0.000] | ... | [1.804,3.215] | [1.189,1.682] |
| Pollution control | [1.565,4.274] | [0.000,0.000] | [0.000,0.000] | ... | [2.515,3.357] | [1.189,1.682] |
| Green image | [1.189,1.682] | [1.091,1.542] | [0.000,0.000] | ... | [1.477,1.915] | [1.044,1.354] |
| Health and safety | [0.000,0.000] | [0.000,0.000] | [0.000,0.000] | ... | [0.000,0.000] | [2.051,2.388] |
| Contractual stakeholder influence | [1.477,1.915] | [1.044,1.354] | [1.044,1.354] | ... | [2.000,2.000] | [1.477,1.915] |
| Local community influence | [0.000,0.000] | [0.000,0.000] | [0.000,0.000] | ... | [0.000,0.000] | [1.185,2.060] |
| Social responsibility management system | [1.477,1.915] | [1.044,1.354] | [0.000,0.000] | ... | [1.477,1.915] | [0.000,0.000] |

Step 3: Determine the total important impact matrix.

Step 3.1: Use Equations (17) and (18) to calculate the rough direct importance influence matrix $D$ and Formulas (19)–(21) to carry out normalization to obtain the normalized rough direct importance influence matrix $C$, as shown in Table 5.

**Table 5.** The normalized rough direct influence matrix.

| | Costs | Quality | Delivery Reliability | ... | Local Community Influence | Social Responsibility Management System |
|---|---|---|---|---|---|---|
| Costs | [0.000,0.000] | [0.023, 0.030] | [0.024, 0.041] | ... | [0.000, 0.000] | [0.000, 0.000] |
| Quality | [0.058, 0.084] | [0.000, 0.000] | [0.000, 0.000] | ... | [0.000, 0.000] | [0.000, 0.000] |
| Delivery reliability | [0.013, 0.020] | [0.000, 0.000] | [0.000, 0.000] | ... | [0.000, 0.000] | [0.000, 0.000] |
| Technology capability | [0.009, 0.013] | [0.012, 0.017] | [0.008, 0.011] | ... | [0.000, 0.000] | [0.000, 0.000] |
| Service | [0.000, 0.000] | [0.000, 0.000] | [0.010, 0.018] | ... | [0.000, 0.000] | [0.000, 0.000] |
| Financial situation | [0.000, 0.000] | [0.000, 0.000] | [0.006, 0.008] | ... | [0.014, 0.031] | [0.000, 0.000] |
| Pollution production | [0.014, 0.022] | [0.000, 0.000] | [0.000, 0.000] | ... | [0.120, 0.149] | [0.016, 0.021] |
| Environmental management system | [0.023, 0.027] | [0.000, 0.000] | [0.000, 0.000] | ... | [0.075, 0.119] | [0.014, 0.025] |
| Green product | [0.012, 0.025] | [0.004, 0.006] | [0.000, 0.000] | ... | [0.030, 0.069] | [0.007, 0.009] |
| Pollution control | [0.009, 0.024] | [0.000, 0.000] | [0.000, 0.000] | ... | [0.070, 0.109] | [0.006, 0.009] |
| Green image | [0.004, 0.006] | [0.003, 0.004] | [0.000, 0.000] | ... | [0.016, 0.021] | [0.004, 0.006] |
| Health and safety | [0.000, 0.000] | [0.000, 0.000] | [0.000, 0.000] | ... | [0.000, 0.000] | [0.014, 0.022] |
| Contractual stakeholder influence | [0.004, 0.006] | [0.003, 0.004] | [0.004, 0.005] | ... | [0.023, 0.030] | [0.005, 0.008] |
| Local community influence | [0.000, 0.000] | [0.000, 0.000] | [0.000, 0.000] | ... | [0.000, 0.000] | [0.004, 0.008] |
| Social responsibility management system | [0.010, 0.018] | [0.006, 0.008] | [0.000, 0.000] | ... | [0.050, 0.073] | [0.000, 0.000] |

Step 3.2: Use Formulas (22)–(24) to calculate the rough total importance influence matrix $T$, and then use Formulas (25)–(30) to convert it to the definite value shown in Table 6.

**Table 6.** The total importance influence matrix in the form of definite value.

| | Costs | Quality | Delivery Reliability | ... | Local Community Influence | Social Responsibility Management System |
|---|---|---|---|---|---|---|
| Costs | 0.006 | 0.032 | 0.040 | ... | 0.021 | 0.003 |
| Quality | 0.084 | 0.002 | 0.002 | ... | 0.016 | 0.003 |
| Delivery reliability | 0.015 | 0.001 | 0.001 | ... | 0.004 | 0.001 |
| Technology capability | 0.014 | 0.015 | 0.010 | ... | 0.011 | 0.001 |
| Service | 0.000 | 0.000 | 0.013 | ... | 0.001 | 0.000 |
| Financial situation | 0.001 | 0.000 | 0.007 | ... | 0.021 | 0.001 |
| Pollution production | 0.022 | 0.002 | 0.002 | ... | 0.180 | 0.027 |
| Environmental management system | 0.031 | 0.002 | 0.002 | ... | 0.140 | 0.029 |
| Green product | 0.019 | 0.006 | 0.001 | ... | 0.058 | 0.010 |
| Pollution control | 0.016 | 0.001 | 0.001 | ... | 0.110 | 0.010 |
| Green image | 0.005 | 0.003 | 0.000 | ... | 0.023 | 0.005 |
| Health and safety | 0.001 | 0.000 | 0.001 | ... | 0.006 | 0.020 |
| Contractual stakeholder influence | 0.006 | 0.003 | 0.004 | ... | 0.030 | 0.007 |
| Local community influence | 0.001 | 0.000 | 0.000 | ... | 0.004 | 0.006 |
| Social responsibility management system | 0.017 | 0.008 | 0.001 | ... | 0.084 | 0.004 |

Step 3.3: Use Formulas (31)–(34) to calculate the sum of rows $x_i$ and the sum of columns $y_i$ of the total importance influence matrix $T$; then, use Equations (38)–(39) to calculate "Prominence" $P_i$ and "Relation" $R_i$.

Step 3.4: Determine the comprehensive weight of the criteria. Use Formula (35) to obtain the comprehensive weight of each evaluation criterion, as shown in Table 7. As shown in Table 7, the result of DEMATEL method can not only calculate the index weight, but also reveal the important influencing factors and internal structure by analyzing the logical relationship between the factors in the system and the direct influence matrix to calculate the influence degree, affected degree, cause degree and centrality of the factors. The analysis process of this method is intuitive and clear. We may need to feedback the evaluation results to the suppliers when we evaluate the suppliers, so as to facilitate the suppliers to improve and further strengthen the cooperation relationship. When the supplier management is faced with many standards to be improved, the best solution is to find the index that has the greatest impact on other criteria. The result of DEMATEL method can well meet this requirement.

**Table 7.** The weights of the criteria.

| | $x_i$ | $y_i$ | $P_i$ | $R_i$ | $w_i$ | Rank |
|---|---|---|---|---|---|---|
| Costs | 0.854 | 0.238 | 1.092 | 0.616 | 0.088 | 5 |
| Quality | 0.561 | 0.077 | 0.638 | 0.484 | 0.056 | 10 |
| Delivery reliability | 0.229 | 0.084 | 0.313 | 0.145 | 0.024 | 14 |
| Technology capability | 0.350 | 0.254 | 0.604 | 0.095 | 0.043 | 12 |
| Service | 0.067 | 0.176 | 0.243 | −0.109 | 0.019 | 15 |
| Financial situation | 0.140 | 0.506 | 0.647 | −0.366 | 0.052 | 11 |
| Pollution production | 0.937 | 0.329 | 1.265 | 0.608 | 0.099 | 2 |
| Environmental management system | 0.961 | 0.253 | 1.215 | 0.708 | 0.099 | 2 |
| Green product | 0.384 | 0.653 | 1.038 | −0.269 | 0.075 | 7 |
| Pollution control | 0.488 | 0.635 | 1.122 | −0.147 | 0.079 | 6 |
| Green image | 0.161 | 0.891 | 1.052 | −0.730 | 0.090 | 4 |
| Health and safety | 0.222 | 0.204 | 0.425 | 0.018 | 0.030 | 13 |
| Contractual stakeholder influence | 0.165 | 1.117 | 1.282 | −0.952 | 0.112 | 1 |
| Local community influence | 0.113 | 0.709 | 0.822 | −0.596 | 0.071 | 8 |
| Social responsibility management system | 0.620 | 0.126 | 0.747 | 0.494 | 0.063 | 9 |

*4.2. Evaluating the Suppliers by FVIKOR*

Four experts evaluated the company's four alternative suppliers, and the evaluation results are shown in Table 8. By using Formulas (36)–(50), the suppliers are sorted, and the results are shown in Table 9. Table 9 ranks the weights of the evaluation criteria selected by sustainable suppliers. Adding the index weight of each dimension in Table 10 shows that the environmental performance index is the most important index dimension (0.442), followed by the economic performance index (0.282) and social performance index (0.276). In addition, Table 10 shows that the most important economic performance index is cost (0.088), followed by quality (0.056), financial status (0.052), technical level (0.043), timeliness of delivery (0.024) and service (0.019). In previous studies, quality was the most important indicator of economic performance; however, the weight of the cost index is greater than that of the quality index in this study. The reason is that, considering the relationship between the cost indicator and other indicators, the $P$ value of the cost indicator (1.092) reveals that the interaction between the cost indicator and the other indicators is very great, far greater than the $P$ value of the quality indicator (0.638). Therefore, the weight of the cost index is greater than that of the quality criterion.

**Table 8.** Experts rate the evaluation of each supplier.

| Expert 1. | Costs | Quality | Delivery Reliability | Local Community Influence | Social Responsibility Management System |
|---|---|---|---|---|---|
| S1 | 1, 2 | 3 | 4 | 2, 3 | 3 |
| S2 | 2, 3, 4 | 4 | 3, 4 | 3 | 3, 4, 5 |
| S3 | 5, 6 | 4, 5 | 5 | 4, 5 | 5 |
| S4 | 3 | 5, 6 | 2, 3 | 4 | 5, 6 |
| **Expert 2** | **Costs** | **Quality** | **Delivery Reliability** | **Local Community Influence** | **Social Responsibility Management System** |
| S1 | 2, 3 | 3, 4 | 3, 4 | 3 | 3, 4 |
| S2 | 3, 4 | 3 | 3, 4 | 3 | 4, 5 |
| S3 | 5 | 4, 5 | 5 | 4 | 6 |
| S4 | 3, 4 | 4, 5, 6 | 3, 4 | 4, 5 | 5, 6 |
| **Expert 3** | **Costs** | **Quality** | **Delivery Reliability** | **Local Community Influence** | **Social Responsibility Management System** |
| S1 | 1, 2, 3 | 3, 4 | 3, 4 | 1, 2, 3 | 3, 4 |
| S2 | 2, 3 | 3, 4 | 3 | 3 | 4, 5 |
| S3 | 2, 3 | 5 | 5, 6 | 4, 5 | 5 |
| S4 | 2, 3 | 4, 5 | 2, 3 | 4, 5 | 5, 6 |
| **Expert 4** | **Costs** | **Quality** | **Delivery Reliability** | **Local Community Influence** | **Social Responsibility Management System** |
| S1 | 2, 3 | 4 | 3 | 2, 3 | 3, 4 |
| S2 | 4 | 3, 4 | 4 | 3 | 3, 4 |
| S3 | 4, 5, 6 | 4 | 5, 6 | 4, 5 | 6 |
| S4 | 4 | 4, 5 | 2, 3, 4 | 4, 5 | 5, 6 |

**Table 9.** Ranking of the alternatives according to values of *S*, *R*, and *Q*.

| Alternatives | *S* | Rank | *R* | Rank | *Q* | Rank |
|---|---|---|---|---|---|---|
| *S1* | 0.702 | 4 | 0.104 | 4 | 1.000 | 4 |
| *S2* | 0.628 | 3 | 0.090 | 3 | 0.743 | 3 |
| *S3* | 0.263 | 1 | 0.071 | 2 | 0.091 | 2 |
| *S4* | 0.300 | 2 | 0.064 | 1 | 0.042 | 1 |

**Table 10.** The different weights of the criteria between the proposed method and AHP-VIKOR.

| Dimension of SSS | Criteria of SSS | The Proposed Method | | AHP-VIKOR | |
|---|---|---|---|---|---|
| | | $w_i$ | Rank | $w_i$ | Rank |
| Eco | Costs | 0.088 | 5 | 0.092 | 4 |
| | Quality | 0.056 | 10 | 0.118 | 3 |
| | Delivery reliability | 0.024 | 14 | 0.067 | 7 |
| | Technology capability | 0.043 | 12 | 0.036 | 12 |
| | Service | 0.019 | 15 | 0.048 | 10 |
| | Financial situation | 0.052 | 11 | 0.025 | 13 |
| Env | Pollution production | 0.099 | 2 | 0.133 | 1 |
| | Environmental management system | 0.099 | 2 | 0.132 | 2 |
| | Green product | 0.075 | 7 | 0.060 | 8 |
| | Pollution control | 0.079 | 6 | 0.079 | 5 |
| | Green image | 0.090 | 4 | 0.039 | 11 |
| Soc | Health and safety | 0.030 | 13 | 0.060 | 8 |
| | Contractual stakeholder influence | 0.112 | 1 | 0.022 | 14 |
| | Local community influence | 0.071 | 8 | 0.020 | 15 |
| | Social responsibility management system | 0.063 | 9 | 0.068 | 6 |

The environmental performance indicators are all very high, with pollutant emissions (0.099), environmental management system (0.099) and green image (0.090) being the most important indicators, followed by pollution control (0.079) and green production (0.075). Because of the implementation of the national environmental protection policy, the requirements for the environmental protection measures of enterprises are high. As the *P* value of the environmental performance index indicates, the *P* values of pollutant emissions (1.265), environmental management system (1.215), green image (1.052), pollution control (1.122) and green production (1.038) are all very high. Thus, the environmental dimension index strongly constrains the economic dimension index and the social dimension index and strongly influences the indicators of these two dimensions. Therefore, the environmental performance index is the most important of the three index dimensions.

The social performance index is the least important of the three index dimensions, but its stakeholder rights and interests index is the most important of all the criteria; this finding is quite different from previous index weight results. The *P* value (1.282) of the stakeholder equity criterion indicates that the overall influence value of this criterion is very high, but the *x* value (0.165) and *y* value (1.117) of this criterion show that the degree of influence exerted by this criterion on other criteria is very small and that it is greatly influenced by other criteria. The *R* value ($-0.952$) of the stakeholder equity criterion also shows that the criterion is the affected party. Many studies have found that the rights and interests of stakeholders are closely related to the cost and quality of products, as well as the pollutant emissions and environmental protection measures of enterprises, and are closely correlated with other criteria [44,45].

The final ranking of the alternatives is in descending order of *S*, *R*, and *Q* values. *S* represents a positive ideal solution, *R* represents a negative ideal solution, and *Q* represents a compromise solution. Therefore, the decision can be achieved according to the descending order of the compromise solution values. As shown in Table 9, scheme *S4* is considered to be the optimal scheme (compromise solution) according to the *Q* value, and the scheme is finally ranked as *S4* > *S3* > *S2* > *S1*. In addition, the *Q* values of schemes *S3* and *S4* are very close, and *S3* can be used as an alternative.

## 5. Comparisons with Other Existing Methods

To illustrate the effectiveness and scientific quality of the proposed method, we compared it with the AHP-VIKOR method that has commonly been used in previous studies [10]. First, use the AHP method to calculate the supplier criteria weight; then, use the VIKOR method to sort the alternative suppliers. The comparative results of the criteria weights are shown in Table 10. The supplier ranking results by AHP-VIKOR are shown in Table 11.

**Table 11.** Ranking of the alternatives.

| Alternatives | *S* | **Rank** | *R* | **Rank** | *Q* | **Rank** |
|:---:|:---:|:---:|:---:|:---:|:---:|:---:|
| S1 | 0.736 | 4 | 0.124 | 4 | 1.000 | 4 |
| S2 | 0.575 | 3 | 0.092 | 3 | 0.439 | 3 |
| S3 | 0.289 | 1 | 0.085 | 2 | 0.028 | 1 |
| S4 | 0.336 | 2 | 0.083 | 1 | 0.053 | 2 |

For the comparative analysis with the results of the AHP-VIKOR method, Figure 2 also shows that the results of the criteria weights obtained by the rough DEMATEL method and AHP method are quite different. In addition, the criteria ranking is also quite different. Except for the environmental management system (ranked 2nd) and the technical level (ranked 12th), the criteria are ranked differently. However, environmental criteria rank high in the results obtained by both methods. The AHP method considers pollutant emissions, environmental management systems and quality to be the three most important criteria. The method proposed in this paper considers stakeholder rights, pollutant emissions and environmental management systems to be the first three most important criteria. The

results show that environmental performance criteria have a significant impact on the evaluation index system of sustainable supplier selection. In addition, as mentioned above, the stakeholder equity index in the rough DEMATEL method ranks highest, while the AHP method ranks 14th; these results are quite different. The local community impact ranks 8th in the rough DEMATEL method and 15th in the AHP method, mainly because the AHP method assumes that the criteria are independent of each other and ignores the problem of criteria correlation. However, the rough DEMATEL method adopted in this paper takes into account the mutual influence relationship between the criteria and makes the determination of the criterion weights more scientific and reasonable. This method was adopted in this paper precisely because of the interaction between indicators, which makes social dimension indicators play a greater role. Considering the influence of social dimension indicators on economic and environmental dimension indicators, the ranking of the two indicators of stakeholder rights and influence on local communities is higher than that of the AHP method. The delivery reliability index ranks 7th in the AHP method, 14th in the rough DEMATEL method; the quality index ranks 3rd in the AHP method and 10th in the rough DEMATEL method. This is because the impact of the quality and delivery reliability criteria on other environmental and social dimension criteria is small, resulting in a low overall ranking of these two indicators.

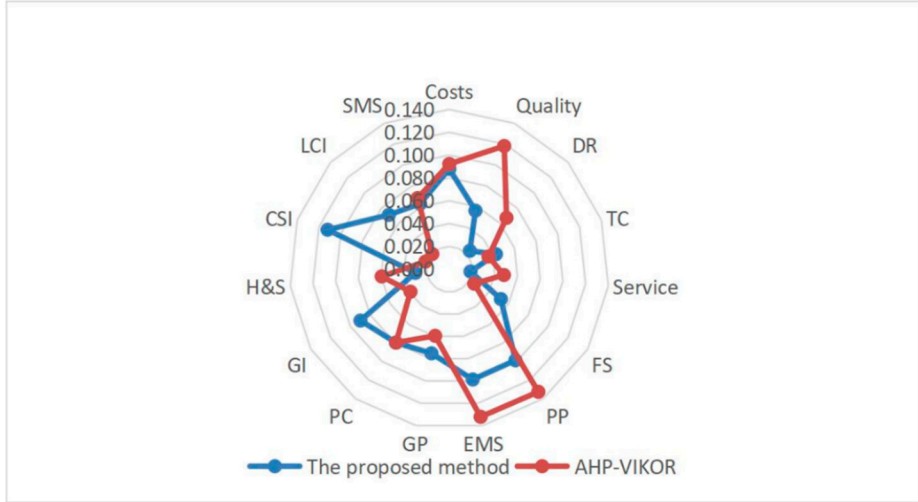

**Figure 2.** Sustainable supplier selection criteria weights derived from the pairwise comparison.

The ranking results of schemes obtained according to AHP-VIKOR are also different, the supplier ranking calculated by the proposed method in this paper is $S4 > S3 > S2 > S1$, and the supplier ranking calculated by the AHP-VIKOR method is $S3 > S4 > S2 > S1$. The method provider $S4$ proposed in this paper ranks first, and $S3$ ranks second, whereas in the AHP-VIKOR method, supplier $S3$ ranks first, $S4$ ranks second, and other suppliers rank the same. This result is mainly due to the difference in index weight, which causes the experts to have different effects on the evaluation results of various suppliers. The stakeholder equity index ranks 1st in the proposed method and 14th in the AHP method; green image ranks 4th in the proposed method and 11th in the AHP method, which is the method used in this case. The indicators played a greater role than in the AHP-VIKOR method, and supplier $S4$ is evaluated more highly in stakeholder equity and green image than supplier $S3$. Therefore, in this case, $S4$ ranks higher than $S3$. The final reason for this result is that the AHP method ignores the interaction between the indicators, whereas the rough DEMATEL method takes it into account. In addition to the importance comparison of the indicators, the impact of the indicators is considered. Therefore, the results of the index weights are different, resulting in differences in the results of the supplier ranking.

## 6. Conclusions

With the deterioration of the environmental situation, an increasing number of enterprises attach importance to the sustainability of supplier selection. A systematic supplier evaluation system can effectively improve enterprise supply chain performance. Sustainable supplier selection research has focused on two aspects: the establishment of a performance evaluation index system and the selection of a supplier evaluation method. Many studies have confirmed that the green supplier evaluation index system considered only traditional and green criteria and ignored social criteria when it was established. As a growing number of experts begin to concentrate on the issue of sustainable supplier evaluation selection, social criteria were gradually included in the consideration of the supplier evaluation index system. This paper not only comprehensively considers three aspects, economy, environment and society, of the sustainable supplier evaluation index system but also provides a scientific model and adopts the rough DEMATEL-VIKOR method to select sustainable suppliers. The rough DEMATEL method is used to determine supplier index weight, and VIKOR is used for sustainable supplier selection. Compared with the AHP-VIKOR method proposed by Luthra et al. [10], the rough DEMATEL method takes into account the interaction between indicators in the process of application, which more clearly shows the important role of environmental indicators and social indicators in the evaluation indicators of sustainable suppliers. The rough DEMATEL method proposed by Song et al. [8] considered the interaction between the criteria in the process of use, but the interaction between the criteria was based mainly on the subjective judgment of evaluation experts, and in-depth analysis is lacking in the literature. In this paper, the mutual influence among the criteria is considered more comprehensively and scientifically in the process of use, and the criterion relation influence table is based on the literature analysis, which is of greater reference value to the sustainable supplier evaluation system. The article uses FVIKOR to evaluate suppliers' choices, considering not only the ambiguity of external information but also that the evaluation process is simple for managers.

The focus of this study is the supplier sequencing problem, which can be used for single/multiple supplier sequencing, but the order allocation problem for multiple suppliers is not considered. In addition, this paper ignores the relationship between manufacturers and suppliers in constructing a sustainable supplier selection evaluation index system and does not consider the impact of the evaluation results on suppliers. In future research, the relationship between manufacturers and suppliers will be considered, enabling a supplier to adjust and improve its own deficiencies according to feedback from the manufacturer. At the same time, the issue of order allocation between multiple suppliers can be further considered.

**Author Contributions:** Both authors contributed equally in the writing of this article. J.Z. and D.Y. designed the whole study, proposed the conceptualization and wrote the original draft; Q.L. conducted modeling and results analysis; B.L. and Y.M. were responsible for project administration and also reviewed and edited the paper. All authors have read and agreed to the published version of the manuscript.

**Funding:** This work was partly supported by National Social Science Foundation of China (No. 18BGL012), Hebei Provincial Higher Education Humanities and Social Sciences Young Top Talent Project (No. BJ2017064), Hebei Natural Science Foundation (No. G2020202008).

**Institutional Review Board Statement:** "Not applicable" for studies not involving humans or animals.

**Informed Consent Statement:** "Not applicable" for studies not involving humans.

**Data Availability Statement:** No new data were created or analyzed in this study. Data sharing is not applicable to this article.

**Conflicts of Interest:** The authors declare no conflict of interest.

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
