# Peer review of "Research on Sustainable Supplier Selection Based on the Rough DEMATEL and FVIKOR Methods"

_sustainability, doi:10.3390/su13010088_

Round 1
Reviewer 1 Report
I do not have any comments and suggestions.
Author Response
Dear Reviewer,
Thank you very much for your recognition for our paper entitled “Research on Sustainable Supplier Selection Based on the Rough DEMATEL and FVIKOR Methods” (ID: sustainability-1025534).
We appreciate for your warm work earnestly. Your recognition is a great encouragement for us, and we will continue to work hard in our future scientific research.

Reviewer 2 Report
The authors propose a new method for evaluating and ranking a sustainable supplier with consideration of mutual correlations between evaluation criteria. To do this, they combine the rough DEMATEL and fuzzy VIKOR methods. The results of the proposed system obtained for a specific numerical example are compared with the existing AHP-VIKOR method.
This is the usual way to introduce a new approach to solving a certain class of practical problems. Unfortunately, in this case, the authors present a very problematic result. First, there is no metric according to which the possible benefits of the proposed method could be objectively assessed.
Secondly - the comparison of the proposed method with the AHP-VIKOR method does not make sense - there are more suitable methods calculating the dependencies in the decision model (fuzzy ANP, fuzzy MOP, system dynamics) that can be used for such a comparison.
Third, the theoretical or practical benefits of the article are unclear - the proposed method is demonstrably neither better nor simpler than existing methods.
Author Response
Dear Reviewer,
Thank you very much for your letter and for the reviewers’ comments concerning our manuscript entitled “Research on Sustainable Supplier Selection Based on the Rough DEMATEL and FVIKOR Methods” (ID: sustainability-1025534). Those comments are considerable valuable and helpful for revising and improving our paper, as well as the important guiding significance to our researches. Based on discussion and research of reviewers’ comments, we have made corrections which we hope meet with approval. Revised portion are marked in red in the paper. The main corrections in the paper and the responds to the reviewer’s comments are as following:
------------------------------------------------------------------------------
C1. Response to comment: This is the usual way to introduce a new approach to solving a certain class of practical problems. Unfortunately, in this case, the authors present a very problematic result. First, there is no metric according to which the possible benefits of the proposed method could be objectively assessed.
Response: Thank you for your valuable advice. We further explain the advantages of the rough DEMATEL and FVIKOR methods proposed in this paper from the method principle and application practice respectively. These instructions are at the first paragraph of the Section 3 and at the end of the section 4.1.
In line 200-210 [see paper 5].
In line 460-470 [see paper 15].
C2. Response to comment: The comparison of the proposed method with the AHP-VIKOR method does not make sense - there are more suitable methods calculating the dependencies in the decision model (fuzzy ANP, fuzzy MOP, system dynamics) that can be used for such a comparison.
Response: In the first paragraph of the Section 3 of the paper, we compare DEMATEL method with ANP method, and explain the reasons why we choose DEMATEL method, and why DEMATEL method is more suitable than ANP method to determine the weight of sustainable supplier evaluation index.
In line 200-210 [see paper 5].
C3. Response to comment: The theoretical or practical benefits of the article are unclear - the proposed method is demonstrably neither better nor simpler than existing methods.
Response: At the end of section 4.1, we supplement the guiding significance of the rough DEMATEL method adopted in this paper in practice. By explaining the actual meaning of some variables such as , , , in the weights of the criteria in table 7, the advantages of the method proposed in this paper are expounded.
For example, is considered as an overall influence of that criterion. The vector reveals the difference between the exerted influence and received influence of the .
Such information is very important for providing feedback information to suppliers in the later stage, which helps suppliers to quickly find out the problems that need to be improved mostly.
In line 460-470 [see paper 15].
In addition, we checked and revised the grammar of the paper sentence by sentence with two native English speakers in the field of Supply Chain Management and Mathematics.
We tried our best to improve the manuscript and made some changes in the manuscript. These changes will not influence the content and framework of the paper. And here we list the changes and marked in rad in revised paper.
We appreciate for your warm work earnestly, because your comments have greatly improved the quality of the paper. And we hope that the correction will meet with approval. Once again, thank you very much for your comments and suggestions.

Reviewer 3 Report
Review of the Manuscript Number: Sustainability-1025534
Title: Research on Sustainable Supplier Selection Based on the Rough DEMATEL and FVIKOR Methods
I would like to thank the authors and editors for having had the opportunity to review this manuscript.
The authors put forward a hybrid approach involving the rough DEMATEL technique and the fuzzy VIKOR (FVIKOR) method for solving the sustainable supplier selection problem. The proposal specifies 15 evaluation criteria and is verified by an appropriate case study. The subject is of interest especially for practitioners and falls well into the scientific profile of the Sustainability journal.
The research is decently settled in the relevant literature. The references are up-to-date, and the presented background is adequate and sufficient. From a substantive point of view, the proposed hybrid methodology is soundly justified and the developed sustainable supplier evaluation index system could be of great importance in decision-making practice. The inclusion of fuzzy methods makes the proposed approach better tailored for the imprecise or linguistic preference statements. I strongly recommend this paper in Sustainability. Below I list minor issues that could further improve the paper.
Minor issues:
(1) I would encourage the author to provide, apart from the general scheme from Figure 1, a more detailed diagram including all steps described in section 3.2. Criteria weight determination method.
(2) Page 3, line 101, the JIT abbreviation is used, but it is nowhere defined. The same with AHP and other acronyms. The authors should revise the manuscript according to this remark.
(3) Page, e.g., 6-7. There is a problem with font sizes in mathematical formulas or variables used in paragraphs. The authors should pay attention to it and correct it if possible.
(4) Page 11, line 420: there is something wrong with the scale; "equally important" is repeated twice in the sentence "... 7-point scale the scores of equally important, equally important, strongly ..."
(5) Page 19, Figure 2, the graph quality is poor, authors should consider exporting it as vector graphics.
Although, the manuscript is, generally, written in quite good English, I have spotted some minor issues in this regard.
(6) Page 1. Two consecutive sentences start exactly the same:
"Multinational companies must use optimal measures ..."
"Multinational companies ..."
The authors should consider rewriting.
(7) Page 3, line 120: Should rather be "Actually, ..." instead of "Acturally, ..."
(8) Page 5, line 207-208: The word method is repeated 3 times in a short sentence: "Both the VIKOR method and the TOPSIS method are common MCDM methods, ...". Consider rewriting.
Author Response
Revision Letter
Dear Reviewer,
Thank you very much for your letter and for the reviewers’ comments concerning our manuscript entitled “Research on Sustainable Supplier Selection Based on the Rough DEMATEL and FVIKOR Methods” (ID: sustainability-1025534). Those comments are considerable valuable and helpful for revising and improving our paper, as well as the important guiding significance to our researches. Based on discussion and research of reviewers’ comments, we have made corrections which we hope meet with approval. Revised portion are marked in red in the paper. The main corrections in the paper and the responds to the reviewer’s comments are as following:
------------------------------------------------------------------------------
C1. Response to comment: I would encourage the author to provide, apart from the general scheme from Figure 1, a more detailed diagram including all steps described in section 3.2. Criteria weight determination method.
Response: We redraw the flow chart of the criteria weight determination method, adding all the steps described in Section 3.2, as shown in Figure 1.
In line 236-238 at the top [see paper 6].
C2. Response to comment: Page 3, line 101, the JIT abbreviation is used, but it is nowhere defined. The same with AHP and other acronyms. The authors should revise the manuscript according to this remark.
Response: We explain the abbreviations of JIT, AHP, DEMATEL, FVIKOR, MCDM, TOPSIS, IT2TrFS, DEA and BWM in the paper. For example, we complement JIT with Just-In-Time (JIT) and AHP is supplemented as analytical hierarchy process (AHP).
[see paper 2-5].
C3. Response to comment: Page, e.g., 6-7. There is a problem with font sizes in mathematical formulas or variables used in paragraphs. The authors should pay attention to it and correct it if possible.
Response: We uniformly adjust the size of the formula in this paper.
C4. Response to comment: Page 11, line 420: there is something wrong with the scale; "equally important" is repeated twice in the sentence "... 7-point scale the scores of equally important, equally important, strongly ..."
Response: We are very sorry for this mistake. We have already referred to "...7-point scale the scores of equally important, equally important, strongly ..."revised to "...7-point scale the scores of equally important, moderately important, strongly ...".
In line 423-425 [see paper 12].
C5. Response to comment: Page 19, Figure 2, the graph quality is poor, authors should consider exporting it as vector graphics.
Response: We have redrawn Figure 2 to make it clearer.
[see paper 19].
C6. Response to comment: Page 1. Two consecutive sentences start exactly the same:
"Multinational companies must use optimal measures ..."
"Multinational companies ..."
The authors should consider rewriting.
Response: Thank you for asking this question, we have rewritten it. We have revised "Multinational companies must use optimal measures..." to "Companies in developed countries must use the optimal measures to select multinational suppliers according to the economic and environmental objectives".
In line 35-37 [see paper 1].
C7. Response to comment: Page 1. Page 3, line 120: Should rather be "Actually, ..." instead of "Acturally, ...".
Response: We have revised “ Acturally, …” to “ Actually, … ”.
In line 118 [see paper 3].
C8. Response to comment: Page 5, line 207-208: The word method is repeated 3 times in a short sentence: "Both the VIKOR method and the TOPSIS method are common MCDM methods, ...". Consider rewriting.
Response: We have rewritten this sentence and now it is revised as “The FVIKOR method is used to select suppliers. Compared with TOPSIS method, this method considers the distance between the decision scheme and the positive ideal solution and the distance from the decision scheme to the negative ideal solution, as well as the relative importance of these distances. [43].”
In line 215-218 [see paper 5].
We tried our best to improve the manuscript and made some changes in the manuscript. These changes will not influence the content and framework of the paper. And here we list the changes and marked in red in revised paper.
We appreciate for your warm work earnestly, because your comments have greatly improved the quality of the paper. And we hope that the correction will meet with approval. Once again, thank you very much for your comments and suggestions.
